# Cytotoxicity of Methacrylate Dental Resins to Human Gingival Fibroblasts

**DOI:** 10.3390/jfb13020056

**Published:** 2022-05-11

**Authors:** Jolanta Sulek, Elzbieta Luczaj-Cepowicz, Grazyna Marczuk-Kolada, Maciej Rosłan, Adam Holownia

**Affiliations:** 1Department of Pharmacology, Medical University of Bialystok, Mickiewicza 2C, 15-222 Bialystok, Poland; jolasulek02@interia.pl (J.S.); roslanmaciek@gmail.com (M.R.); 2Department of Paedodontics, Medical University of Bialystok, Waszyngtona 15A, 15-269 Bialystok, Poland; elzbieta.luczaj@umb.edu.pl (E.L.-C.); grazyna.kolada@umb.edu.pl (G.M.-K.)

**Keywords:** apoptosis, cytotoxicity, HSP70, human gingival fibroblasts, miR-9, methacrylate resins, necrosis, oxidative stress

## Abstract

This study aimed to assess the acute and delayed cytotoxicity of three, popular light-cured methacrylate-based restorative resins (MRs): Charisma (C), Estelite (E), and Filtek (F), to human gingival fibroblasts in culture. Cells were grown for up to 24 h with light-cured (or pre-cured) resins. We evaluated resin cytotoxicity, redox imbalance, necrosis/apoptosis, miR-9, and heat shock protein 70 (HSP70). The role of resin-induced oxidative stress (damage) in HSP70-response (repair) was assessed using binary fluorescence labeling. All MRs decreased viable cell numbers and cell proliferation and damaged cell membranes, and their 24 h-delayed toxicity was lower (C), higher (F), or similar (E) to that induced by freshly-cured resins. Cell membrane damage induced by C and E decreased with time, while F produced a linear increase. All resins generated intracellular oxidative stress with the predominant necrotic outcome, and produced heterogeneous responses in miR-9 and HSP70. The double fluorescence (damage/repair) experiments pointed to common features of E and F but not C. In the subset of cells, the binary response induced by E and F was different from C, similar to each other, and positively interrelated. Experimental data show that selective MR cytotoxicity should be taken into account when considering repetitive use or massive reconstruction.

## 1. Introduction

Methacrylate-based resins (MR) are widely used in everyday dental practice, but accumulating records indicate that their toxicity may be clinically relevant [1,2,3,4]. They can trigger local irritation and an allergic reaction in the dentist and the patients. They frequently cause hypersensitivity, asthmatic reactions, neurological symptoms, irritation, and dermatological reactions. In conservarive dentistry, the published data point to the role of partial polymerization in acute toxicity [4,5,6]. In turn, deleyed toxicity may be related to decomposition of polymerized materials [7].

Chemicals that are used to induce polymerization and to reduce resin shrinkage, especially methacrylates and epoxy compounds, and their metabolites are clearly toxic [4,5]. They affect proximate oral tissues [5,6,7,8]. Some monomers such as triethylene glycol dimethacrylate (TEGDMA) or 2-hydroxylethyl methacrylate (HEMA) are definitely cytotoxic [9,10]. Methacrylate monomers produce reactive oxygen species (ROS), induce apoptosis and/or genotoxicity and affect cell proliferation in culture [11,12]. It was shown that resin-derived compounds induce chromatin condensation, DNA double-strand breaks, and cytotoxicity to human gingival fibroblasts [13,14].

In this study, we used three popular, light-cured MRs: Charisma (C), Estelite Sigma Quick (E), and Filtek Z550 (F). C contains a monomer of toxic bisphenol-A-glycidyl methacrylate (BisGMA) and a highly toxic TEGDMA [15]. E contains BisGMA and TEGDMA, and benefits from free radical-amplified photopolymerization [15], while F is a nanohybrid composite, containing genotoxic urethane dimethacrylate (UDMA) and toxic ethoxylated bisphenol-A methacrylate (BisEMA) [16]. All these compounds induce oxidative stress but can also diffuse to surrounding fluids and exert direct toxicity [17,18]. This study aimed to assess and characterize, at least in part, the acute and delayed cytotoxicity of C, E and F to human gingival fibroblasts in culture. We have used 2D fibroblasts cultures before to study toxicity of dental pulp capping materials [19]. In contrast to 3D cultures, 2D cultures offer more consistent results and are appropriate for studying toxicity [20]. Moreover, we evaluated and quantified apoptosis, oxidative stress and two putative epigenetic and biochemical biomarkers of MR toxicity: miR-9, a stress-related microRNA, which has broad indicative potential as a transition factor of normal fibroblasts into cancer-associated fibroblasts, and is involved in epithelial-mesenchymal transition, wound healing and cancer [21], and a key chaperone and stress marker protein, heat shock protein 70 (HSP70). HSP70 is one of the most abundant molecular chaperones, and has a role in all stages of protein life including oxidative protein damage [22]. Additionally, the mutual relationship between MR-induced oxidative stress and the heat shock response was assessed. 

## 2. Materials and Methods 

In this study, we used Charisma (C), produced by Heraeus GmbH, Germany, Estelite Sigma Quick (E), and a supranano, spherically-filled universal composite commercialized by Tokuyama Dental Corporation, Japan and Filtek Z550 (F) a nanohybrid universal restorative material produced by 3M. The shade of all resins was A3. Table 1 presents their chemical characteristics.

### 2.1. Preparation of Resins

Each material was prepared according to the manufacturer’s instructions. The 5 × 5 mm^2^ cylinders were formed, polymerized on both sides for 2 × 20 s using 1000 mW/cm² —1200 mW/cm² intensity with blue curing light generated by a Led B Woodpecker and placed into cell culture inserts with translucent PET membranes (0.4 µm pore diameter; Grainer Bio-one, Kremsmunster, Austria) physically separating the resins from gingival fibroblasts. Freshly prepared resins were incubated with cells for up to 24 h. To assess delayed toxicity, cured resins were pre-incubated for 24 h in culture media and then they were applied to the cells for 24 h using cell culture inserts as above.

### 2.2. Cell Culture

An adherent immortalized cell line of human gingival fibroblasts (ATCC^®^ CRL-2014HGF-1, f.) was grown in Dulbecco’s Modified Eagle’s Medium containing 10% fetal bovine serum (GIBCO Invitrogen, Grand Island, NE, USA) and 1% antibiotic at 37 °C, 5% CO_2_ and 95% humidity. In each experiment, one resin cylinder was covered with 2 ml of culture medium.

### 2.3. Cytotoxicity and Cell Cycle Assays

MTT (3-[4,5-dimethylthiazol-2-yl]-2,5-diphenyl tetrazolium bromide; Sigma, St. Louis, MO, USA) assay was used with 100,000 cells per well to assess cell viability [23]. The net absorbance (OD570-OD650) of untreated cells was taken as 100% viability. 

Time-dependent toxicity was assessed using lactate dehydrogenase (LDH)-release assay (LDH cytotoxicity kit, ScienCell, Carlsbad, CA, USA) from 500,000 cells.

The cell cycle was quantified by flow cytometry in cells seeded at a density of 50,000 cells per well basing on the quantity of the cellular DNA bound to a fluorescent propidium iodide (PI) [24].

### 2.4. Apoptosis/Necrosis

To quantify apoptotic and necrotic cell numbers, 50,000 cells were double-stained with fluorescein isothiocyanate (FITC)-conjugated annexin V (Clontech Laboratories, Mountain View, CA, USA) and analyzed using a FACScan flow cytometer (FC; Becton Dickinson, Franklin Lakes, NJ, USA). Positive apoptotic controls were prepared using doxorubicin treatment (1 µg/mL), while necrosis was induced by exposing the cells to hyperthermia (15 min at 50 °C) [25].

### 2.5. Oxidative Stress

The generation of reactive oxygen intermediates was analyzed in FC in cells grown with MR for 3 or 6 h and then loaded with 5 μΜ dichloro-dihydrofluorescein diacetate (H_2_DCFDA; Sigma Chem. Co., Poznan, Poland) [26]. For quantification, the fluorescence from 10,000 cells was captured.

### 2.6. HSP70 Expression

HSP70 protein was quantified by Western blots using a specific monoclonal antibody against HSP70 (Abcam, Cambridge, MA, USA; a rabbit monoclonal antibody against human HSP70), a secondary anti-rabbit antibody (Abcam, Cambridge, MA, USA) linked to horseradish peroxidase and enhanced chemiluminescence detection. Blots were quantified using Image Quant software (TL 7.0. General Electric Co, Freiburg, Germany).

Changes in HSP70 expression were also measured in double fluorescence experiments together with DCF to delineate the role of the heat shock response in resin-induced oxidative stress. We addressed the issue by double staining 100,000 cells using rabbit monoclonal antibodies against human HSP70 conjugated to Alexa Fluor 647 (Abcam, Cambridge, MA, USA) and DCFDA. Samples were analyzed in FC. Experimental data were plotted as bivariate cytograms, scatterplots, fluorescence density plots, and histograms and were analyzed for the vector, central tendency and spread using the FlowJo (V10.8.1, Ashland, OR, USA) and Flowing Software (v.2.5.1. freeware, Turku University, Finland).

### 2.7. MiR-9 Assay

To quantify miR-9, reverse transcription and quantitative PCR (qPCR) was performed using the TaqMan™Thermo Fisher Scientific microRNA assay Kit (Waldham, MA, USA). U6 small RNA was used as a reference. miR-9 expression was calculated with the ΔΔCt method and was shown as fold changes.

### 2.8. Statistical Analysis

Statistical analysis was performed by Analysis of Variance (ANOVA), followed by Bonferroni test adjustment for multiple comparisons at a 95% confidence level. 

## 3. Results

### 3.1. MTT Test 

Table 2 and Figure 1 (panel A) show MR cytotoxicity by MTT test. All freshly-cured resins significantly decreased cell viability. The highest toxicity was generated by C (58% decrease in viable cell numbers; *p* < 0.01), while E and F produced 22% (*p* < 0.05) and 31% (*p* < 0.05) declines, respectively. In delayed toxicity experiments, pre-incubated C was significantly less toxic (C’ vs C—*p* < 0.01), F was considerably more toxic (F’ vs F—*p* < 0.01), while E (E’ vs E) exerted similar cytotoxicity.

### 3.2. Cell Cycle

Table 2 and Figure 1 (panel D) show subpopulations of hypodiploid cells, resting, and proliferating cells. All resins were cytotoxic and engendered significant antiproliferative effects. The highest DNA damage was produced by C (almost 5-fold increase in hypodiploid cell numbers vs. control cells; *p* < 0.01), while in cells grown with E or F, the subsets of damaged cells increased by 2 (*p* < 0.01) and 3 (*p* < 0.01) fold, respectively. 

Considering cell proliferation, the greatest inhibitory effect was observed with E (about 60% decrease; *p* < 0.01), an approximately 35% decrease (*p* < 0.01) was detected in cells cultured with F, and about a 17% decline (*p* < 0.05) was induced by C. In addition, C but not E or F decreased by about 40% (*p* < 0.01) the numbers of resting cells. 

### 3.3. Cell Membrane Damage

All MRs produced cell membrane damage (Table 2, Figure 1, panel C). In cells growing with C, increased LDH activity (*p* < 0.05) in the culture medium was already detected after 1 h. Then cell membrane damage increased with time to attain, after 24 h, more than 40% of total enzyme activity; however, the cytotoxicity plot was hyperbolic. The plot of LDH release from fibroblasts incubated with E was similar to C, but the maximal values registered at 24 h were lower (28% of total enzyme activity; *p* < 0.05). F produced a 37% peak value at 24 h, but in contrast to C and E, the time-dependent cytotoxicity plot was linear (correlation coefficient r = 0.99). 

### 3.4. Apoptosis/Necrosis

Figure 2 and Table 2 show the quantification of cell necrosis and apoptosis.

Apoptotic cells bound to annexin V-FITC, but not to PI, are shown in the lower right (4) quadrant of data plots. Necrotic cell staining with both PI and annexin V-FITC appear in the upper left (1) quadrant. Untreated cells were positioned to have 2% annexin V-FITC binding alone, 3% of the population stained with both annexin V and PI and 2% of PI-alone binding. C (panel D) visibly induced necrotic changes. Almost 75% of cells were stained with PI, but not with annexin V (PI^+^A^−^; quadrant 1; panel D). E produced a mixed pattern of changes. About 48% of cells were assigned as necrotic (PI^+^A^−^), while about 28% of cells had both necrotic and apoptotic features (PI^+^A^+^; quadrant 2, panel E). F generated mostly necrotic changes (quadrant 1, panel F), and about 73% of cells were assigned as PI^+^A^−^; however, a substantial number of cells (about 16%) developed both necrotic and apoptotic attributes (quadrant 2, panel F). 

### 3.5. Oxidative Stress

Figure 3 (panel A) shows a typical, right-shifted DCF fluorescence histogram in fibroblasts grown for 3 h with C, and a corresponding, overlaid histogram from control cells. 

Median oxidative stress after 3 h of cell growth with C, E or F is shown as bars in Figure 3 (panel A) and in Table 2. In all samples, a mean DCF fluorescence was significantly increased. The highest increase (by about 12 fold; *p* < 0.01) was observed in cells grown with E, and slightly lower (about 10 fold; *p* < 0.01 and 9 fold; *p* < 0.01) fluorescence intensity was documented in cells grown with C and E, respectively. 

### 3.6. MiR-9 and HSP70

miR-9 (Table 2) was significantly increased in all groups (in C and E groups – *p* < 0.05) with an almost 3.4 fold (*p* < 0.01) increase in cells grown with F and a less than 2 fold increase in C and E. Figure 3 (panel B) shows the expression of HSP70 in fibroblasts grown with C, E or F for 24 h. A significant (about 7 fold; *p* < 0.01) increase in HSP70 protein levels was observed only in cells grown with C.

### 3.7. Binary Scatter Plots of Oxidative Stress and HSP70

Table 2 and Figure 3 (panel C) show mean values and typical binary scatterplots with scatter area, central tendency lines, trend lines, and their slopes reflecting changes of oxidative stress (green DCF fluorescence) and HSP70 expression (red Alexa Fluor 647 fluorescence), respectively. 

MRs produced different binary scatterplots with variable scatter areas, trend and vector lines. Both E and F resulted in a visible narrowing, and prolongation of the fluorescence dispersion plots, which suggests that both variables may be positively interrelated, especially when both values are increased. C resulted in a significant increase in the scatter area and less consistent variables. Corresponding relative area values were 167% (*p* < 0.01), 127% (NS) and 131% (*p* < 0.05) of control, for C, E, and F, respectively. 

Considering the analysis of central lines and trends, vector trend lines and their slopes (Figure 3C), there was about a 20% deviation of the central line slope to DCF over HSP70 in cells grown with C when compared to (similar) slopes of E and F. The slopes of vector lines were the same in E and F but very different (*p* < 0.01) from the downward shifted (C), significantly different from slopes of the central trend line (C; *p* < 0.01 and E; *p* < 0.05) and similar to F. Moreover, graphical data suggest the existence of local trends and at least two cell subsets especially in control cells and in cells grown with E. 

## 4. Discussion

### 4.1. Resin Cytotoxicity

In the present work, we assessed the effect of three popular light-cured MRs on the proliferation and survival of human gingival fibroblasts in culture. The published data point to the major role of partial polymerization in acute toxicity [4,5,6]. Our data indicate that freshly polymerized resins and resins polymerized for 24 h before cell experiments induce significant cytotoxicity. Considering freshly cured resins, the highest cytotoxicity was induced by C, while E and F caused less relevant declines in cell viability. Freshly-cured C was significantly more cytotoxic to gingival fibroblasts than C, preincubated for 24 h prior to cell contact, E produced a similar effect while the delayed cytotoxicity of F was even higher. Our experiments indicate that MR cytotoxicity is important and highly variable. It is possible that resin polymerization may be the most relevant factor in MR toxicity since uncured resins may exert very intense cytotoxicity [27]. It is well known that cells cultured in a monolayer are more sensitive to toxins than 3D cultures [20]. Therefore experimental data are model-dependent and general conclusions should be cautious.

#### Delayed Cytotoxicity and Cell Cycle

In vitro experiments indicated that polymerized methacrylate resins can release similar amounts of toxic compounds for at least 74 h [28]. According to published data, BisGMA has the highest toxicity to human gingival fibroblasts compared to gradually less toxic UDMA, TEGDMA, and HEMA [6,29,30]. Considering chemical profiles, both C and E contain BisGMA and TEGDMA, but both materials exert different toxicity, possibly due to a CQ, which is used in C as a photo-initiator [31]. To measure cytotoxicity, we began with a very popular MTT method, however, the values of the MTT test are strongly affected by cell numbers. To discriminate between cytotoxic and antiproliferative effects, changes in cell cycle were analyzed. It turned out that apart from cytotoxicity, all resins engendered significant antiproliferative effects. The strongest inhibition of cell growth was caused by E, while F and C produced lesser, but still significant declines. C affected mostly non-dividing cell numbers while E and F primarily decreased fractions of proliferating cells. Relatively high cytotoxicity of C was not linked to relevant growth inhibition, while E, and to a lesser extent F, exerted significant antiproliferative effects, but only limited cytotoxicity. Antiproliferative action of methacrylates was also described in human gingival fibroblasts [32,33,34], and this effect should be taken into account while testing resin cytotoxicity in cell lines. Published data emphasize inconsistent results of MTT and LDH-release tests in experiments of resin cytotoxicity in human gingival fibroblasts. The MTT test was described as a more sensitive assay than the LDH test [35], but it seems that inhibition of cell proliferation was not taken into account. This effect is especially important in testing long-term cytotoxicity in proliferating cells.

To further characterize cell–MR interactions we did time-dependent LDH-release experiments. The highest cytotoxicity at 24 h was observed in fibroblasts grown with C. In our experiments, the highest cytotoxicity at 24 h was again obtained with C, and the time-dependent cytotoxicity plot was hyperbolic. The corresponding plot for E was also hyperbolic with a lower plateau than the C-plot, whereas the cytotoxicity of F grew linearly. Our data show that experiments designed to determine mechanisms of time-dependent toxicity may be relevant to assess long term toxicity and resin biocompatibility. 

### 4.2. Oxidative Stress

All MR produced oxidative stress, which was not surprising since all materials benefit from free radical-mediated reactions during polymerization [36,37]. The highest levels of oxidative stress were produced by the significantly less toxic F, but not by the highly toxic C or E. It seems that intracellular oxidative stress is not a single and specific element in the noxious interaction of dental resins with fibroblasts. 

### 4.3. Necrosis and Apoptosis

MR produced cell death via necrosis or apoptosis to specific cell subpopulations. The pathways of cell death are of key importance to the diagnostics of resin-induced cytotoxicity since they are relevant in elaborating prevention strategies. In our study, C and F produced predominantly nonspecific necrotic changes, while E generated a mixed pattern of alterations with some apoptotic-like changes. Literature data on TEGDMA and HEMA-induced apoptosis are inconsistent [10,15], and it seems that the relative presence of the apoptosis should be confirmed by intracellular apoptotic markers. Our results indicate that it is hard to predict the specific pathway of resin-induced cell death using published methacrylate toxicity data. Consequently, the properties of each particular dental material should be carefully examined irrespective of its chemical profile. 

### 4.4. MiR-9 Expression

Cell growth, apoptosis, differentiation, and survival is regulated in part by small endo- or exogenous non-coding miRNAs that bind to the 3′-untranslated regions of the target mRNAs and alter transcription of several target genes relevant to cell physiology and pathology. The changes in expression of miR-9 in our experiments were significant, but not very high. It was shown that miR-9 silencing can inhibit cell death induced by hydrogen peroxide [38] while increased miR-9 was observed in epithelial cells subjected to epithelial-mesenchymal transition, which is interrelated to cancer [39]. The role of miR-9 in dental resin toxicity is not known, but considering mutagenic proprieties of methacrylates [6] and important roles of non-coding miRNAs in cell growth, apoptosis, differentiation and survival [40], it seems to be crucial to include more epigenetic markers into a systematic study of MR toxicity.

### 4.5. HSP70 Expression and HSP70/DCF Assay

We have shown, that C significantly increases HSP70 levels in gingival fibroblasts, while changes in HSP70 levels in cells grown with E or F are not significant. HSP70 may protect cells from thermal, chemical or oxidative stress by helping refold the damaged proteins and inhibit apoptosis [41]. It was shown that both HEMA and TEGDMA suppress HSP72 expression in human monocytes [42]. Increased HSP70 levels in fibroblasts grown with C indicate that the protein is induced and/or stabilized in response to toxic stimuli but it is not able to abolish cell damage and fibroblasts proceed to necrosis. The double fluorescence experiments support the above statements but point to some heterogeneity of cell responses to MR. It seems that in control cells there are at least two distinct cell subsets with different HSP70 to oxidative stress ratios. In cells with the highest oxidative stress (e.g., some cells grown with F), HSP70 levels are also highly elevated and it is possible, that at least in a fraction of affected cells, both parameters are interrelated. It seems that MRs affect both redox cell balance and exert more distinct chemical reactions, which share some similarities in cells grown with E and F. 

### 4.6. Conclusion

Our in vitro experiments have focused on adverse effects of popular dental MR to human gingival fibroblasts in culture. However, given their diverse effects and possible selective “affinity” to some cells, it is too early at this stage, maybe apart from avoiding skin contact to the selected material, to propose better clinical practices. It seems, that neither HSP70 nor miR-9 can be used as a universal marker of MR cytotoxicity. However, considering the variable chemical profiles and inconstant, possibly transient pattern of changes induced by MR-based resins to the cells, it seems that in the preclinical testing of heterogeneous materials, the sequential analysis of single-cell behavior and subsequent analysis of functional trends within selected cell subpopulations may help to build a better algorithm to highlight resin-specific features.

## Figures and Tables

**Figure 1 jfb-13-00056-f001:**
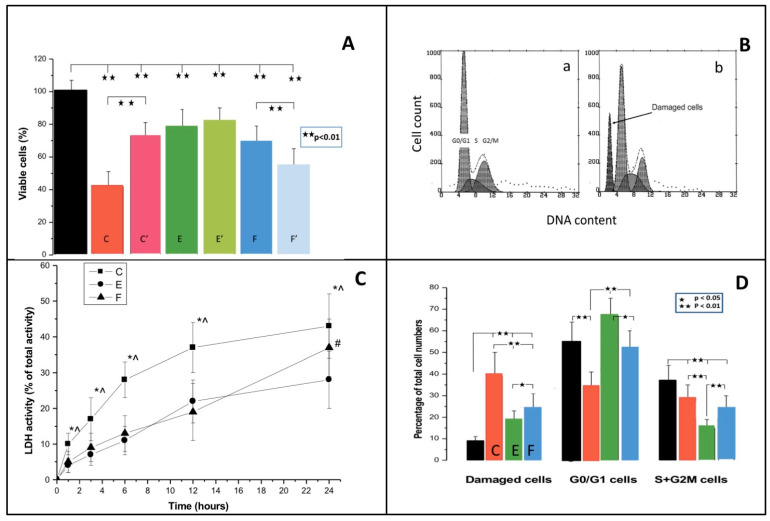
The effect of Charisma (C), Estelite (E), and Filtek (F) on the viability of human gingival fibroblasts is shown in panel (**A**). Cells were grown for 24 h with freshly cured dental resins (C, E, F) or with dental resins preincubated for 24 h (C′, E′, F′), and then applied to the cells. Cell viability was measured with MTT test and compared with 100% viable, naïve cells (n = 10). Panel (**B**) shows two typical histograms of propidium iodide-DNA fluorescence of control cells (a), cells co-cultured for 24 h with E (b), and bars reflecting cytotoxicity and cell cycle analysis. Cells were grown with dental resins for 24 h, stained with propidium iodide, and analyzed by flow cytometry. Original histograms were obtained using FACS Canto II flow cytometer, gating was set for the control cells and was applied to other experimental samples. Cell distribution (panel (**D**)) was quantified using MultiCycle software as “subdiploid” (damaged cells), diploid (G0/G1peak)-pre-DNA synthesis/resting and proliferating cells (S-phase-DNA synthesis plus G2/M-post-DNA-synthesis/mitotic cells). Panel (**C**) shows time-dependent cytotoxicity of C, E, and F. Cells were co-cultured with dental resins and lactate dehydrogenase (LDH) activity in the culture media was measured at 1, 3, 6, 12, and 24 h. Total LDH activity was estimated in lysed control cells (*—*p* < 0.05 for comparison with the control cells; #—*p* < 0.05 for comparisons with the cells grown with C; ^—*p* < 0.05 for comparisons with thecells grown with E).

**Figure 2 jfb-13-00056-f002:**
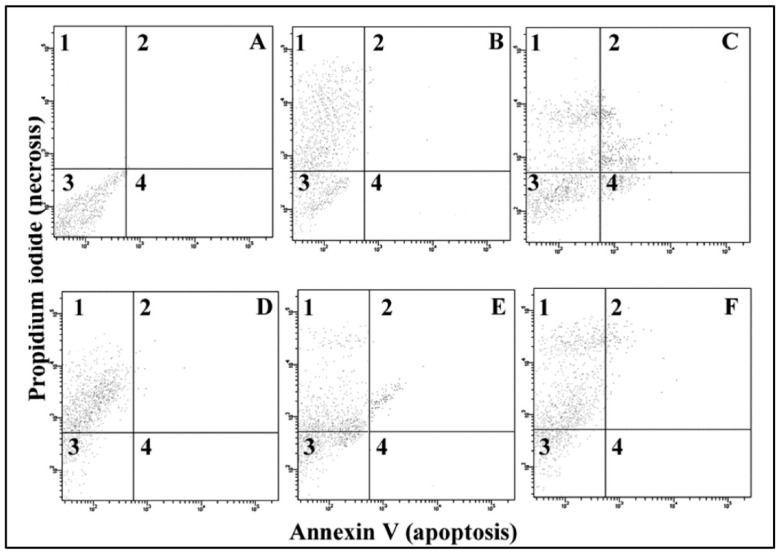
Quantification of apoptosis/necrosis with annexin V/propidium iodide in human gingival fibroblasts grown with dental resin Charisma (C), Estelite (E), or Filtek (F) for 24 h. Part (**A**)-control cells, part (**B**)-necrotic cells (necrosis was induced by fifteen minutes hyperthermia at 50 °C), part (**C**)-apoptotic cells (apoptosis was induced by overnight doxorubicin treatment; 1 µg/mL). Parts (**D**–**F**) are cells grown for 24 h with C, E, and F, respectively. Quadrant 1 represents the proportion of necrotic cells, quadrant 2 shows non-viable apoptotic cells stained with both propidium iodide and annexin V-FITC, quadrant 3 represents naïve cells, while quadrant 4 shows apoptotic cells.

**Figure 3 jfb-13-00056-f003:**
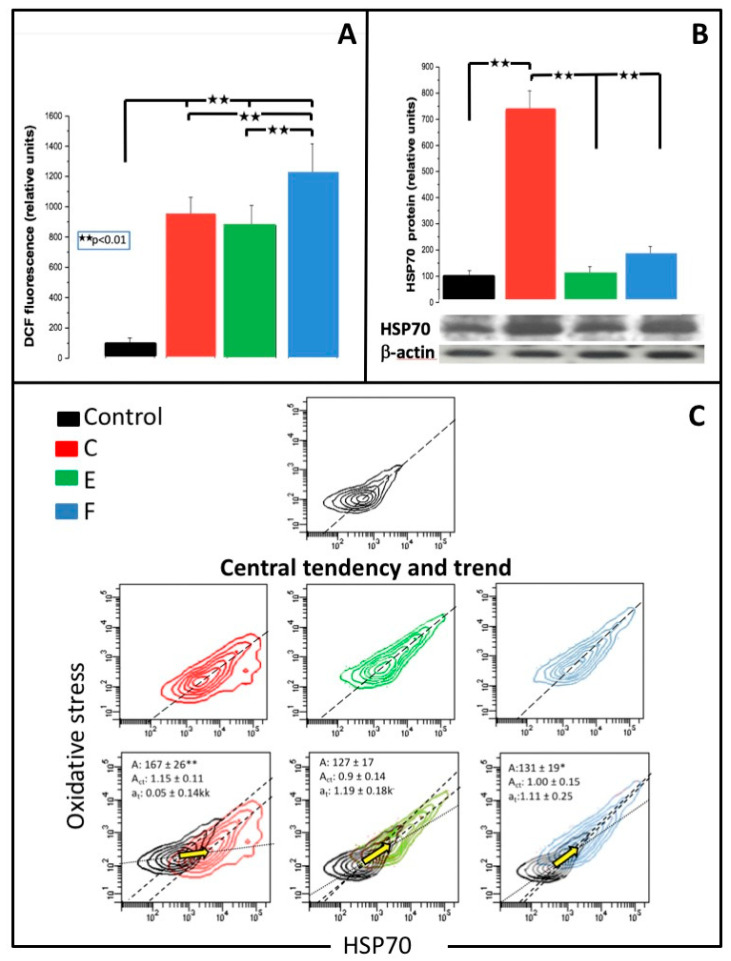
Panel (**A**) shows oxidative stress in human gingival fibroblasts grown for 3 h with light-cured dental resins: Charisma (C), Estelite (E), or Filtek (F). Cells were stained with dichlorofluorescein diacetate (DCFDA) and analyzed by flow cytometry. Typical histograms (overlay) of green DCF fluorescence in naïve human gingival fibroblast and in fibroblasts grown for 3 h with C are included. Panel (**B**) shows the expression of heat shock protein 70 (HSP70) quantified by Western-blot in cells grown for 24 h with C, E, or F. Panel (**C**) shows binary scatterplots of oxidative stress (green DCF fluorescence; relative units; damage) vs. HSP70 expression (red Alexa Fluor 647 fluorescence; relative units; repair) in cells grown for 6 h with C, E, or F. Moreover scatter area (A), central tendency lines (------) and their slopes (A_ct_), and trend lines (.......) with vectors and their slopes (a_t_) are displayed.

**Table 1 jfb-13-00056-t001:** Chemical constituents of Charisma (C), Estelite Sigma Quick (E), and Filtek Z550 (F) composite resins.

Resin	Chemical Constituents
C	Bisphenol-A-glycidyl methacrylate (BisGMA); Triethylene glycol dimethacrylate (TEGDMA); Camphorquinone (CQ)
E	Bisphenol-A-glycidyl methacrylate (BisGMA); Triethylene glycol dimethacrylate (TEGDMA)
F	Urethane dimethacrylate (UDMA); Bisphenol-A- polyethethylene glycol diether dimethacrylate (BisEMA)

**Table 2 jfb-13-00056-t002:** Cytotoxicity (MTT test) of freshly cured and pre-cured (pre-incubated) resins, time-dependent cell membrane damage (LDH release), cell cycle (PI-DNA assay), oxidative stress (DCF fluorescence), induction of apoptosis/necrosis (annexin V-FITC/PI), expression of heat shock protein 70 (HSP70; WB), miR-9 expression (RT-PCR) and quantification of oxidative stress-dependent HSP70 expression and in naïve human gingival fibroblast cells growing for 3 h (DCF), 6 h (DCF/HSP70 scatterplots) or 24 h (all other experiments) with methacrylate dental resin Charisma (C), Estelite (E), and Filtek Z550 (F).

Biochemical Indices of Resin Toxicity	Control	C	E	F	Significance
MTT test (% of control)					
Freshly-cured resin	100 ± 7	42 ± 9 **	78 ± 11 **##	69 ± 10 **##	Toxicity C >> E = F
Preincubated resin		73 ± 8 **++	81 ± 9 **##	54 ± 11 **##^^	F > E > E
LDH (% of total activity)					
1 h		10 ± 3 *	4 ± 2	5 ± 3 *	C—Rapid increase, saturable
3 h		17 ± 6 **	7 ± 3 **	9 ± 4 **	E—Slow increase, saturable
6 h		28 ± 5 **	11 ± 4 **	13 ± 5 **	F—Slow increase, linear
12 h		37 ± 7 **	22 ± 6 **	19 ± 8 **	
24 h		43 ± 9 **	28 ± 8 **	37 ± 8 **	
Cell cycle (%)					
Cells with damaged DNA	8 ± 3	39 ± 11 **	18 ± 5 **##	24 ± 7 **##^	Increased C > F > E
Pre-DNA synthesis/ resting	55 ± 9	34 ± 7 **	67 ± 8 ##	52 ± 9 ##^	Variable E > C = F
Proliferation	37 ± 7	27 ± 8 **	15 ± 4 **##	24 ± 6 **^^	Decreased E > F > C
Oxidative stress (relative units)	100 ± 34	951 ± 111 **	877 ± 132 **	1215 ± 201 **##^^	F > E = C
Apoptosis/Necrosis (%)					
Naive cells	93 ± 7	18 ± 6 **	21 ± 11 **	20 ± 5 **	C-predominantly necrosis
Necrotic cells	2 ± 3	75 ± 10 **	48 ± 14 **#	73 ± 11 **	
Apoptotic cells	2 ± 2	2 ± 2	3 ± 2	1 ± 2	D - some apoptosis
Necrotic/apoptotic cells	3 ± 2	5 ± 4	28 ± 13 **#	16 ± 7 **##^^	F-mostly necrosis
miR-9 (fold change)	1	1.76 *	1.92 *	3.39 *	Increased F > C = E
HSP70 (relative units)	100 ± 17	733 ± 76 **	112 ± 23 ##	177 ± 28 ##	Increased C >> (E = F normal)
DCF/HSP70 (relative units)					
Scatterplot area	100 ± 14	167 ± 26 **	127 ± 17	131 ± 19 *	Increased C > E = F
Central tendency line (slope a_ct_)	1.04 ± 0.12	1.15 ± 0.11	0.9 ± 0.14	1.00 ± 0.15	
Trend (vector) line (slope a_t_)	-	0.05 ± 0.14 κκ	1.19 ± 0.18 κ^−^	1.11 ± 0.25	C ≠ E = F

* *p* < 0.05; ** *p* < 0.01 for comparisons with the control cells. # *p* < 0.05; ## *p* < 0.01 for comparisons with the cells grown with C. ^ *p* < 0.05; ^^ *p* < 0.01 for comparisons with the cells grown with E. ++ *p* < 0.01 for comparisons with the cells grown with the same freshly-cured resin. κ *p* < 0.05; κκ *p* < 0.01 for comparisons with corresponding a_ct._

## Data Availability

Not applicable.

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
