# Peer review of "Cytotoxicity of Methacrylate Dental Resins to Human Gingival Fibroblasts"

_jfb, 2022, doi:10.3390/jfb13020056_

Round 1
Reviewer 1 Report
The article jfb-1677012 deals with a biological characterization of the human fibroblasts response in contact with resins applied as fresh light-cured or aged pre-cured materials. A deep characterization has been provided bringing to the conclusion that materials are unfortunately responsible for the release of toxic elements prohibiting for cells’ proliferation by genetic damage.
The work is of interest as the toxicity (with a special emphasis to the late toxicity due to degradation) of dental materials is still a significant topic in Dentistry where sometimes the economic aspects are predominant to the experimental evidences. Experiments are well conducted, and results are convincing and properly discussed in light of the previous literature. Therefore, I can suggest this manuscript for publication after some minor modifications aimed at improving the discussion. Here my comments:
Major comments:
Materials’ aging: to simulate a cumulative release of potential toxic chemicals from the resins, the Authors pre-submerged specimens for 24 hs in the culture medium. This preliminary step seems to significantly affect the results (by both increasing or decreasing toxicity) thus resulting a crucial aspect to take into account. However, the aging protocol must be better discussed and justified in light of the selected materials to explain if the selected method can resemble materials’ degradation. In my opinion, 24 hs is a short time for materials like resins as well as the medium does not look like the best environment to force degradation that usually requires to be boosted by an acidic pH (resembling saliva) or the presence of enzymes speeding up deterioration. This section must be better discussed and compared to previous literature.
Cells cultivation: despite the use of cells monolayer in presence of culture inserts releasing potential toxic elements is an acceptable method to simulate a non-direct toxic effect, it is nowadays well accepted that cells in 2D are much more sensitive to toxic elements than 3D models resembling the complexity of the tissue. Therefore, the correct choice of the culture model can bring to different results. So, taken into account the complexity of these 3D models and the high costs of the commercial equivalents, the deep analysis provided by the Authors can be acceptable at this stage but this issue must be taken into account in the Discussion as future evaluations performed with the suggested models can bring the Authors to different conclusions.
Minor comments:
abstract: “red-ox” is probably redox
chapter 2.2: I suggest substituting “permanent” with “immortalized”. Please specify the number of cells seeded for the test.
Author Response
Comment 1: “Materials’ aging”
Response: Thank you for this suggestion. It would have been interesting to explore this aspect. However, in the case of our study, it seems slightly out of scope. Material aging or decomposition, although important, was not covered by our algorithm, and actually we decided to focus on acute toxicity. There is no published data on toxicity of the same methacrylate resins. Pre-submerged resins were used in LDH release experiment only. Otherwise we did not focus on time-dependent toxicity. Examination of resin degradation may be interesting regarding accumulative toxicity but probably not in cell culture, where long-term incubation is not possible. Use of human primary cells may be more appropriate for this purpose. We have added additional explanation to the Introduction and Discussion and References.
Comment 2: “Cells cultivation”
Response: We agree. 2D and 3D models are different. Unlike 2D, 3D models can mimic tissue structure and function and provide more accurate data about cell-to-cell interactions. However, in 2D models the effect of complex toxic material are easier to assess and is more target focused. We believe that 3D is superior in drug assays when the major mechanism and molecular pathways are generally known. Briefly, 2D seem to be more convenient in toxicology, especially for scanning, 3D is much more appropriate in pharmacology. Considering multicomponent dental materials and variability of expected effects 2D model is more approprate. We have incorporated your suggestion throughout the manuscript with 1 recent review article on 2D and 3D models and a reference to our earlier work.
Minor comments:
Response: abstract: “red-ox” is now redox throughout the manuscript
Response: chapter 2.2: “Permanent” was substituted with “immortalized” and information on cell numbers per experiment was added in each method.
Reviewer 2 Report
- There were some mistakes in Abstract like “three, popular light-cured 8 methacrylate-based (MR) restorative resins”. Authors should carefully check the manuscript.
- In clinical application, light-cured resinused to fill subgingival cavities are in contact with the gingiva, but mainly with gingival epithelial cells rather than fibroblasts. So, authors should clarify the clinical significance of this study in Introduction section.
- The Introduction section was too easy. Authors only introduced the background of Methacrylate-based resins and cytotoxicity of monomers. The effects of cytotoxicity of methacrylate resins on dental practice rather than “epithelial-mesenchymal transition, wound healing and cancer”should be added in the Introduction section.
- There were some mistakes in the figures like the missing x-axis in histogram in Figure 1A,B and Figure 3A
- The data in the Table 2 was repeated with the content in the following Figures and should be simplified.
- Mechanism research was too simplistic. Why authors quantified the expression level of miR-9 and HSP70? There haven’t been enough reason. Which pathway is involved?
In general, the content of the article is not in-depth enough, the clinical significance is unclear, and major revisions are required.

Author Response
Response 1: The MS was carefully checked and some mistakes were corrected. The indicated sequence in the Abstract i.e. “three, popular light-cured 8 methacrylate-based (MR) restorative resins” is correct. 8 is a line number added by the JFB journal.
Response 2. You have raised an important point here. In vitro cytotoxicity models are indicative, but cannot be translated to clinical effect. When multicomponent composite (dental) materials are tested it is even more difficult. However, such models are commonly used, because in simple cellular models it is possible to point out at specific targets related to specific material and quantify dose-response and analize time-dependent toxicity curves.
Epithelial cells are forming protective layers and their half-life is usually short. They easily die and are naturally replaced by new cells. Fibroblasts are more resistant, structural cells thus potential damage may have log-lasting consequences. They are widely used in toxicity study (approx 2000 papers in PubMed) .
Response 3. Thank you for this suggestion. It would have been interesting to explore this aspect. However, in the case of our study, it seems slightly out of scope. Please note that we did experimental work in the field of biochemical toxicology. Therefore, in the Introduction, we mentioned "epithelial-mesenchymal transition, wound healing and cancer” and we focussed less on clinical aspects, which obviously are the most relevant. Methacrylates are widely used in dentistry. There have been concerns about their potential toxicity, both for the patient and in the workplace. They can cause local irritation of the mucosa and even an allergic reaction. Both the dentist and the patients suffer from hypersensitivity, asthmatic reactions, local neurological symptoms, irritation, and local dermatological reactions. Considering the local effects - the polymerization process is never complete and leakage of unreacted methacrylate monomers occurs during clinical care. The main mechanism is believed to be the ability of monomers to deplete glutathione, thereby causing toxicity. But methacrylate monomers vary in structure and hydrophilicity. It is therefore likely that they may have a different mechanism and dynamics of toxic reactions. We have shown, that the methacrylate resins differ in their ability to induce oxidative stress, and it does not fully correspond to their toxicity. It is important to elucidate the possible mechanisms and we believe we made some relevant steps.
Common and significant clinical problems related to the use of methacrylate in dental practice have now been added to the Introduction, Discussion and Reference sections.
Response 4. X-axes in Figures 1A/B and 3A were intentionally removed. Y ax is numerical, x is not. Bars are numbered so we believe that x ax is not needed.
Response 5: Thank you for this suggestion. Figures in the MS and graphical abstract are now simplified by adding vertical lines and one type of asterisk throughout to show statistical significance. They are much clearer now.
Response 6: There is nothing wrong with simplicity. It is practical and reproducible. A simple test is really needed to assess the toxicity of methacrylate (and possibly other) materials. Moreover, there is nothing practical in PubMed, but instead, a dentist working with methacrylates is encouraged to use the hood to protect themselves and their patients.
Defining pathways (primary in pharmacology) is secondary (in toxicology) since each resin is different and the most toxic should be eliminated and usually there is no specific antidote.
…”Why authors quantified the expression level of miR-9 and HSP70? There haven’t been enough reason”
Response: Well, we believe there were several reasons, but maybe they were not written clearly. We tried to propose a universal marker of metacrylate resin toxicity and experimentally confirm its utility in a binary detecting system. Such a molecule should have a broad-spectrum, possibly linear response to toxins, sensitivity, specificity to the type of cell damage, and be easily quantifiable. The heat shock protein 70 was carefully chosen. The chaperone is responsible for proteostasis, play essential role in protein folding, disaggregation, and degradation of proteins damaged by oxidized proteins. It is one of the most abundant molecular chaperones, has a role in all stages of protein life. Morover Hsp70 is highly inducible. In our experiments Hsp70 was almost perfect, although the response to different resins was variable.
miR-9 emerged as an important regulator in both development and disease. It is easily quantifiable, has certainly broad indicative potential as a transition factor of normal fibroblasts into cancer-associated fibroblasts, involved in extracellular matrix remodelling, apoptosis, cell growth, and steroid resistance.
W were mostly interested in its possible role in cells transition. I turned out that miR-9 is not really highly induced by resins and we did not follow this pathway. Instead in this manuscript, we proposed a simple, but perhaps universal system to polarize the cells and map individual cell (or fraction of cells) or cell behaviour. To our knowledge nobody did it before. Certainly, it is fully arbitrary and can be still ameliorated.
Which pathway is involved?
Response: Most probably oxidative protein damage, certainly damage of cell membrane and destruction of selected cells mostly by necrosis.
Reviewer 3 Report
The manuscript was relatively easy to read in general. However, I would suggest that Table 2 should have a new column presenting the significance of Bonferroni test. Instead of giving multiple symbols at the end of each data cell, something like "F > C = E > Control", "F > C, and E > control" will be more readible.
Author Response
Response:
Thank you for pointing this out. Now a new column was added to Table 1 with more readable comparisons. Such a simplified form was already used in the graphical abstract. Now multiple symbols (on Figures and Graphical abstract ) are also substituted by vertical bars to show the statistical significance with * or ** symbols. Figures are now much clearer.